# What's all that racket! Soundscapes, phenology, and biodiversity in estuaries

**Agnieszka Monczak**[1,2☉], **Bradshaw McKinney**[1], **Claire Mueller**[1], **Eric W. Montie**[1☉]*

**1** Department of Natural Sciences, University of South Carolina Beaufort, Bluffton, South Carolina, United States of America, **2** Institute of Biological and Environmental Sciences, University of Aberdeen, Aberdeen, United Kingdom

☉ These authors contributed equally to this work.
* emontie@uscb.edu

**Data Availability Statement:** All relevant data are within the paper and its Supporting Information files. The raw wav files are not part of the minimal data set but can be made available upon request.

## Abstract

There is now clear evidence that climate change affects terrestrial and marine ecosystems and can cause phenological shifts in behavior. Utilizing sound to demonstrate phenology is gaining popularity in terrestrial environments. In marine ecosystems, this technique is yet to be used due to a lack of multiyear datasets. Our study demonstrates soundscape phenology in an estuary using a six-year dataset. In this study, we showed that an increase in acoustic activity of snapping shrimp and certain fish species occurred earlier in years with warmer springs. In addition, we combined passive acoustics and traditional sampling methods (seines) and detected positive relationships between temporal patterns of the soundscape and biodiversity. This study shows that passive acoustics can provide information on the ecological response of estuaries to climate variability.

## Introduction

Over the past decades, researchers have reported changes in the cyclic nature of migratory and breeding patterns in fauna (e.g. insects, amphibians, and birds) associated with climate change [1–3]. It is important to monitor these shifts because the response of individual species may vary and could disrupt interactions with other species, leading to ecosystem imbalance (e.g. trophic mismatch) [4, 5]. Current advances in passive acoustic monitoring allow for long-term assessments of soundscapes, which can provide instrumental information on biological processes [6]. Long-term monitoring of sound can provide information on the timing of recurring phenomena (e.g. migration, foraging, and spawning) and can detect shifts in these biological processes [7, 8]. In marine ecosystems, there seems to be a knowledge gap in understanding soundscape phenology, simply because there are few long-term datasets.

Soundscape ecology is evolving with novel technological advances, providing alternative approaches in assessing behavior as well as community structure, function, and dynamics [6]. Recently, there has been an increasing interest in assessing whether passive acoustics can capture terrestrial biodiversity [e.g. 9, 10]. Applications to marine ecosystems are more challenging but evidence suggests that sound diversity can reflect species diversity (i.e. measured by underwater visual census of fish) in mangrove, coral reef, seagrass, and rocky habitats [11, 12].

**Funding:** This work was supported by Spring Island Trust, Town of Bluffton/Beaufort County, multiple University of South Carolina (USC) ASPIRE internal awards, Community Foundation of the Lowcountry, SC EPSCoR/IDeA Program award (#17-RE02), South Carolina Aquarium, Research Initiative for Summer Engagement (RISE) grant from USC, The LowCountry Institute, USCB Sea Islands Institute, Palmetto Bluff Conservancy, and the Port Royal Sound Foundation. This work was also supported, in part, by the Southeast Coastal Ocean Observing Regional Association (SECOORA) with NOAA financial assistance award number NA16NOS0120028. The statements, findings, conclusions, and recommendations are those of the author(s) and do not necessarily reflect the views of SECOORA or NOAA. All funds were awarded to EWM. The funders had no role in study design, data collection and analysis, decision to publish, or preparation of the manuscript.

**Competing interests:** The authors have declared that no competing interests exist.

In the southeast USA, significant contributors of biological sound to estuarine ecosystems include snapping shrimp (genus *Alpheus* and *Synalpheus*) and soniferous fish (families Batrachoididae and Sciaenidae). Snapping shrimp produce short, broadband calls using their claw, that have been mainly associated with territorial interactions, communication, and foraging. Soniferous fish species including silver perch *Bairdiella chrysoura*, black drum *Pogonias cromis*, spotted seatrout *Cynoscion nebulosus*, and red drum *Sciaenops ocellatus* produce species-specific calls by rapidly moving a pair of sonic muscles against their swim bladder [e.g. 13–17]. Captive studies have shown that calls produced by these fish species are mostly associated with courtship behavior and reproduction [e.g. 16 & 17]. Many sound-producing species use estuaries periodically for spawning and nurseries, and sound / species richness can vary seasonally [18, 19].

In this study, we deployed three passive acoustic recorders for six years (February 2013—December 2018) in the May River, South Carolina (SC), USA. In addition, we used a catch and release method (i.e. haul seines) to assess species diversity and abundance in intertidal creeks located in close proximity to recording platforms. The specific objectives were to: (i) determine temporal patterns of high, low, and broadband frequency sound pressure levels (SPLs) over the six year time span; (ii) determine how certain environmental factors influence SPLs; (iii) examine phenology of acoustic activity of snapping shrimp and sound producing fish species (i.e. measured as changes in high and low SPLs, respectively); and (iv) determine temporal patterns of species diversity and abundance, and examine how these indices correlate with the soundscape.

## Results

### Temporal patterns of the estuarine soundscape

Comparisons of high (7,000–40,000 Hz), low (50–1,200 Hz), and broadband (1–40,000 Hz) frequency SPLs from 2013 to 2018 revealed temporal and spatial differences (Figs 1 and 2, S1 Fig). Broadband analysis included all physical sounds, biological calls and vocalizations, and anthropogenic noise. Low frequency SPLs included fish calls, the lower bandwidth of snapping shrimp snaps, bottlenose dolphin *Tursiops truncatus* vocalizations (which were few and random), physical sounds, and anthropogenic noise. High frequency SPLs included snapping shrimp snaps, high frequency vocalizations of bottlenose dolphins, physical sounds, and anthropogenic noise.

We observed distinct temporal patterns in SPLs that were influenced by location, year, lunar phase, day/night, tidal phase, temperature, day length, and rainfall. From the three random forest models tested, the models including temperature explained the most variance in the data as compared to models that included rainfall or day length as a factor (S1 and S2 Tables). The designed models with temperature explained 88%, 54%, and 68% of the data variability for high, low, and broadband SPL, respectively (S1 Table). We applied the same models to the data that excluded files with physical sounds and anthropogenic noise (i.e. biological sounds) for the high, low, and broadband frequency analysis. We observed similar results (i.e. models that included temperature as a factor explained 90%, 56%, and 71% of the data variability for high, low, and broadband SPL, respectively) (S2 Table). The factors that were most significant in influencing SPL were location, temperature, and year (S1 and S2 Tables). We observed significant differences in SPL values of biological origin among stations with 14M having the highest values and 9M the lowest (p < 0.05). We detected the highest contribution of anthropogenic noise (i.e. recreational boats) at station 37M (i.e. near the Intracoastal Waterway; 11% of files analyzed) and the lowest at station 9M (i.e. near the headwaters; 2% of files analyzed) (S2–S4 Figs). The anthropogenic noise detected was the most prevalent during the day in the summer months (S2–S4 Figs).

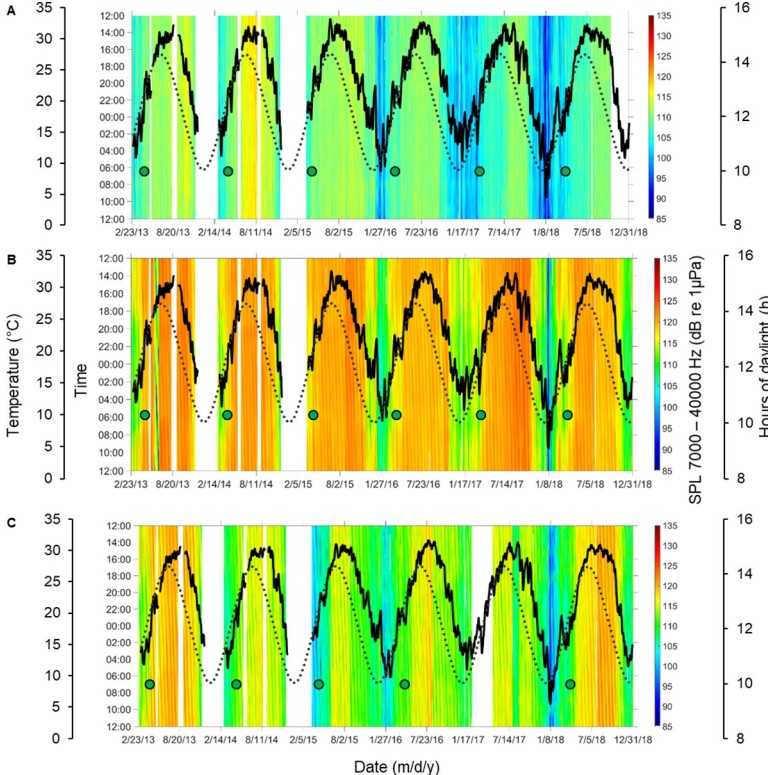

**Fig 1. Time series of high frequency Sound Pressure Levels (SPL) from 2013 to 2018.** Heat maps represent temporal and spatial patterns of high (i.e. 7000–40,000 Hz) frequency SPLs reflecting snapping shrimp acoustic activity at stations (A) 9M, (B) 14M, and (C) 37M in the May River, SC. Time is shown between noon and noon of the next day. Gaps in data = white, temperature = black line, and daylight hours = dotted line. Green dots indicate first posterior probability (PP) of change $\geq$ 0.5 detected during springtime. At station (C) 37M first PP was not calculated for spring 2017 due to missing acoustic data. This dataset contained files with physical sounds and anthropogenic noise.

One striking pattern that we observed was in the seasonal fluctuations of SPLs, which increased and decreased with the seasonal temperature changes of the estuary (Figs 1 and 2 and S1 Fig). These patterns (and the results presented below) were preserved even when files with physical sounds and anthropogenic noise were removed, indicating that the sound patterns were biological in nature (S2–S4 Figs). With an increase of water temperature for every 0.5°C, 0.6°C, and 0.8°C during the springtime, we detected a corresponding increase of 1 dB in SPL for high, low, and broadband frequencies, respectively. Higher values of SPLs were present in the summer months as compared to lower values in the winter, early spring, and late fall (Figs 1 and 2 and S1–S4 Figs). We observed significant differences in SPLs among years for all examined frequency ranges ($p < 0.05$). The highest SPL values occurred in 2013 and 2014, the lowest in 2016 and 2017. In addition, snapping shrimp acoustic activity (i.e. measured as SPLs in the high frequency bandwidth of 7,000–40,000 Hz) was higher during the day, low tide, and new moon as compared to the night, high tide, and full moon ($p < 0.05$). SPLs within the low frequency band were higher during the night (i.e. associated with fish chorusing) and followed an oscillating pattern associated with the lunar phase with higher values recorded on the first quarter of the lunar phase ($p < 0.05$). Values in the broadband SPL frequency range, which reflected a combination of all biological sounds, were the highest during the night, new moon and falling tide ($p < 0.05$).

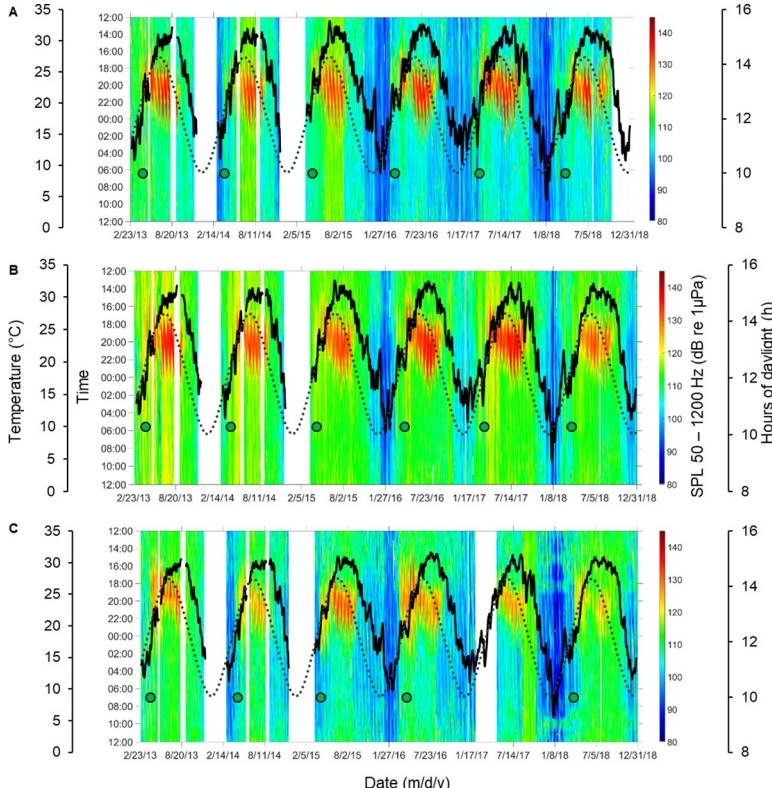

**Fig 2. Time series of low frequency Sound Pressure Levels (SPLs) from 2013 until 2018.** Heat maps represent temporal and spatial patterns of low (i.e. 50–1200 Hz) frequency SPLs reflecting fish and lower frequency range of snapping shrimp acoustic activity at stations (A) 9M, (B) 14M, and (C) 37M in the May River, SC. Time is shown between noon and noon of the next day. Gaps in data = white, temperature = black line, and daylight hours = dotted line. Green dots indicate first posterior probability (PP) of change $\geq 0.5$ detected during springtime. At station (C) 37M first PP was not calculated for spring 2017 due to missing acoustic data. This dataset contained files with physical sounds and anthropogenic noise.

## Soundscape phenology

Based on six years of data recorded at three stations, we examined the phenology of acoustic activity of snapping shrimp (as a measure of high frequency SPL) and fish (as a measure of low frequency SPL) by detecting the date of the first abrupt change in SPL (i.e. posterior probability or PP $\geq 0.5$). We calculated the first abrupt change for both data sets (i.e. one with all sounds and noise, and the data set that excluded physical sounds and anthropogenic noise) but report exact dates from the pure biological dataset (Fig 3; S3 and S4 Tables; S5–S13 Figs). In spring recordings at station 9M, we found the first peak in high frequency SPL to occur on April 9, April 3, April 1, March 25, March 23, and March 29 in 2013, 2014, 2015, 2016, 2017, and 2018, respectively (S3 Table; S2 and S5 Figs). At the same station, we detected the first peak in low frequency SPL to occur on April 2, March 27, April 6, March 25, March 25, and March 28 in 2013, 2014, 2015, 2016, 2017, and 2018, respectively (S4 Table; S3 and S6 Figs). Similar patterns were found at stations 14M and 37M (S3 and S4 Tables). We found that in years with higher mean spring water temperatures, the first peak in high, low, and broadband SPL occurred earlier as compared to years with lower mean spring water temperatures (Fig 3A–3C). During the spring of 2017, an increase in acoustic activity of snapping shrimp was detected 8 days earlier than the 6-year average at station 9M (S3 Table). At the same station, during spring of 2017, acoustic activity of soniferous fish was detected 4 days earlier than the 6-year average (S4 Table). Mean water temperature

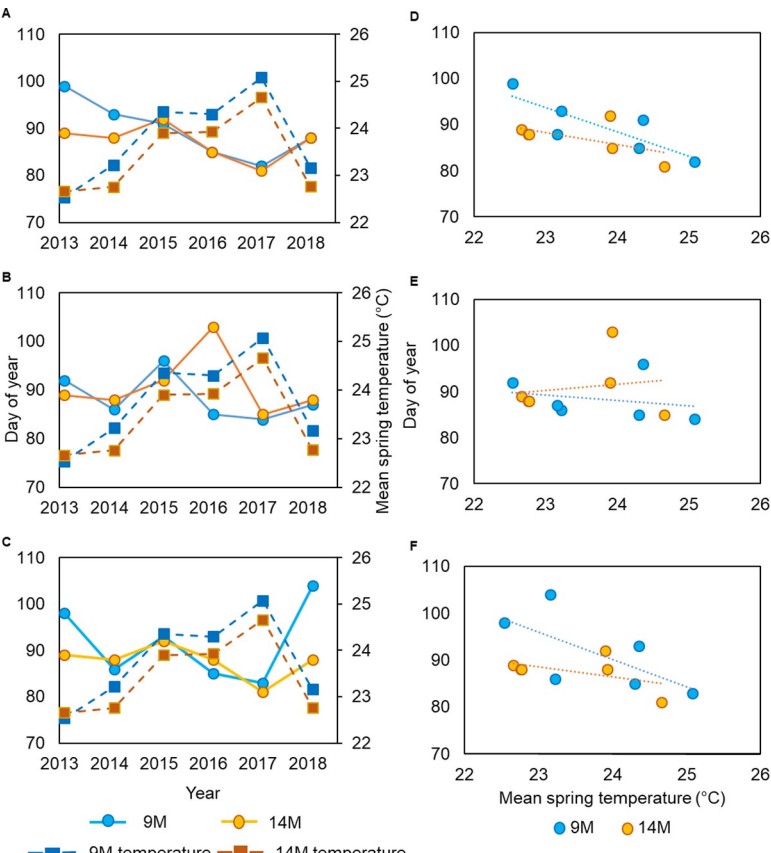

**Fig 3. Relationship between mean spring water temperature and day of year of first Posterior Probability (PP) of change ≥ 0.5.** Left panels: lines with dots represent day of year of first PP ≥ 0.5 of sound pressure level in (A) high (7000–40,000 Hz), (B) low (50–1200 Hz), and (C) broadband (1–40,000 Hz) frequency range, while lines with squares represent mean spring water temperature at stations 9M and 14M. Right panels: relationship between mean spring water temperature and day of year of first PP ≥ 0.5 of sound pressure level in (D) high (7000–40,000 Hz), (E) low (50–1200 Hz), and (F) broadband (1–40,000 Hz) frequency range with corresponding mean spring water temperature at stations 9M and 14M. This dataset did not contain files with physical sounds and anthropogenic noise.

during spring of 2017 was the highest (i.e. + 1.21˚C than the 6 year average) of all 6 years monitored (S3 Table). On the other hand, the mean spring temperature in 2013 was the lowest (i.e. -1.33˚C than the 6 year average), and the first peak in snapping shrimp acoustic activity was detected 9 days later than the 6 year average at station 9M, while fish acoustic activity was detected 4 days later. We found similar patterns at stations 14M and 37M (S3 and S4 Tables). We found negative correlations between mean spring water temperature and the timing of the first peak in probability of change for high, low, and broadband SPLs at stations 9M and 14M (Fig 3D–3F). In addition, during the winter of 2017–2018, we recorded the lowest, minimum water temperature and the lowest SPL values of all three winters monitored (Figs 1 and 2). During the spring 2018, the first peak in SPL was detected later than in the years with higher winter and spring temperatures (S3 and S4 Tables). In the years with higher fluctuations in spring water temperature, there were more abrupt changes in SPLs (S5–S13 Figs).

## Soundscape and biodiversity

We used haul seines to estimate species richness, the Shannon-Wiener diversity index, and total abundance of species in the May River estuary between 2016 and 2018. In total, we caught

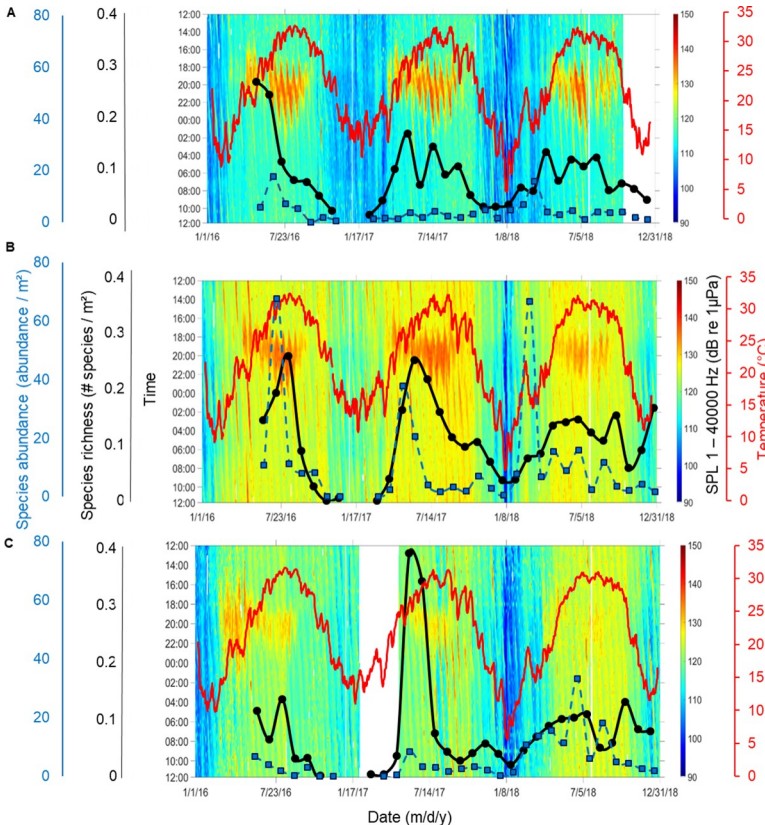

**Fig 4. Time series of broadband frequency SPLs, species richness, and species abundance from 2016 until 2018.**
Heat maps represent temporal and spatial patterns of broadband (1–40,000 Hz) frequency SPLs reflecting all biological
sounds with corresponding species richness (black line), species abundance (blue dash line), and temperature (red
line) at stations (A) 9M, (B) 14M, and (C) 37M in the May River. Gaps in data = white. This dataset contained files
with physical sounds and anthropogenic noise.

5 species of invertebrates and 54 species of fish during seining of which some of these species
are capable of producing sound (S5 Table). However, many of the fish species caught in the
seines were young-of-the-year, and sound production of these juveniles is unknown (S5
Table). We detected temporal patterns in species richness, the Shannon-Wiener diversity
index, and abundance for invertebrates and fish species. We found lower species diversity and
abundance during wintertime (i.e. cooler season), and higher species diversity and abundance
during spring and summertime (warmer seasons). This temporal pattern of species diversity
and abundance followed the warming and cooling patterns of the estuary as well as the oscillat-
ing pattern of the biological soundscape (Fig 4 and S14 Fig). We found significant positive cor-
relations between species richness (as well as the Shannon-Wiener diversity index and
abundance) with high, low, and broadband SPLs (S15A–S15C Fig, S15E–S15G Fig and S16A–
S16C Fig); the highest correlations occurred with low frequency SPL (S15B and S15F Fig). In
addition, we found a significant positive regression between water temperature and species
richness (as well as Shannon-Wiener diversity index) but not between temperature and species
abundance (S15D and S15H Fig, and S16D Fig). Spatially, overall years, the highest species
richness, Shannon-Wiener Diversity index, and species abundance occurred at station 14M,
where we observed the highest SPL values (Fig 4 and S14 Fig).

## Discussion

In this study, we used a six-year passive acoustic dataset to understand the annual and inter-annual variability of an estuarine soundscape. Our findings show a strong relationship between temporal changes in acoustic activities of estuarine organisms and environmental factors. We showed that the transition between winter and spring is a dynamic time-period with an increase in biological sound during the spring, which mirrors the increase in phytoplankton, zooplankton, invertebrates, and fish abundance that drive changes in primary, secondary, and tertiary productivity within estuaries [20, 21]. In years with warmer spring temperatures, this seasonal transition occurred earlier than in years with cooler spring temperatures. This means that temperature plays an important factor in initiating certain behaviors (e.g. spawning), and earlier occurrences of these behaviors reflect an organismal response to climate variability [8].

### Soundscapes and limitations

In addition to sounds of biological origin (e.g. snapping shrimp and fish sounds), factors such as physical and anthropogenic sounds, water depth, and bottom topography may affect received SPLs. Water flow, rain, wind, or wave action, unlike sounds of biological origin, occur randomly [22–25]. These sounds are dominant in the low (200–2000 Hz) and high (15–20 kHz) frequency bandwidths [23]. However, during our analysis, we identified and removed acoustic files that contained sounds associated with intense water flow, rain, wind/wave action, and anthropogenic origin. In addition, this tidal estuary is subject to less wind and wave action as compared to open ocean environments. Thus, we are quite confident that the SPLs and patterns presented are of biological nature. Studies have shown that water depth and active space can affect sound propagation [24, 25]. In our studies, water depth and river width increased from the source towards the mouth and could potentially affect received SPL measurements.

### Soundscape phenology

In our study, we showed that sound production in snapping shrimp and sound producing fish species could serve as potential indicators of climate driven changes in spring phenology. In fact, snapping shrimp respond very quickly to changes in temperature with increased snapping activity with warmer temperatures and decreased activity with cooler temperatures [26]. Warmer temperatures also have the potential to impact spawning phenology of certain fish species that have temperature-dependent gonadal development [27]. Our studies have shown that positive temperature anomalies increase sound production in fish, while negative temperature anomalies decrease calling [28]. In many fish species, spawning seasons are temperature dependent since biologically important processes (e.g. maturation of gonads) require specific temperature ranges. Similar to other studies, we detected a significant increase in snapping shrimp and fish acoustic activity in the spring and summer and a significant decrease in the fall and winter [e.g. 20, 21, 26, 29, 30]. These temporal variations in biological sound levels indicate that there is a strong connection between sound production and the seasonal changes in estuarine diversity and productivity.

In terrestrial ecosystems, the influence of climate change on phenology is well documented and significant; however, in marine environments, this aspect is understudied due to the inability to sample at the necessary time scales [4, 5, 8, 31, 32]. It is important to note that monitoring soundscapes can assist in climate and phenology studies. By tracking vocalizations of amphibians and birds, there is now clear evidence that climate influences the phenology of breeding and migratory patterns [33, 34]. For example, in a terrestrial system, Buxton et al. (2016) detected shifts in songbird phenology in varied thrush (*Ixoreus naevius*), Pacific wren

(*Troglodytes pacificus*), and ruby-crowned kinglet (*Regulus calendula*) due to an earlier winter to spring transition at Glacier Bay National Park, Alaska, USA [8]. Studies have shown that common frogs (*Rana temporaria*) tend to breed earlier in warmer ponds, while 78 songbird species in North America shifted their spring arrival earlier from overwintering grounds due to rising spring temperatures [33, 34]. In marine ecosystems, passive acoustics offers an autonomous, technology-based approach to track spawning behaviors and migratory arrivals of species that produce sounds, which is particularly useful in underwater habitats where visibility can be limited (e.g. estuaries) and access can be challenging (e.g. deep ocean). With the advent of newer, affordable recording systems and increased computational power, underwater sound data can be collected and visualized at short time intervals (e.g. continuously, 20 min, or 60 min) and over long-term scales (i.e. years and decades) providing excellent temporal coverage to detect changes in phenology.

## Soundscape and biodiversity

Many of the sound-producing fish species collected in haul seines were young-of-the-year, and the ability of this life stage to produce sound is questionable. In the May River estuary, adult male oyster toadfish, silver perch, spotted seatrout, and red drum are the major sound-producing species that produce courtship calls and choruses associated with spawning [16, 17, 18, 28]. All of these fish to some degree contribute to sound pressure levels in the low frequency bandwidth [28]. In the May River, oyster toadfish, black drum, silver perch, and spotted seatrout are residents in estuaries year-round, while adult red drum may move offshore in the colder winter months; none of these fish produces sound in the cold, winter months from November to February [18, 28].

Louder habitats may correlate with higher species richness and abundance [11, 12]. In the present study, we showed that higher species diversity and abundance occurred during seasonal periods (i.e. spring and summer) when biological sound levels in the low and high frequency bandwidths were the highest. Furthermore, biologically louder areas of the tidal river had higher diversity and abundance of invertebrates and fish. It is possible that the soundscape could provide organisms with information about habitat quality, resources, and potential predators [6]. It is also possible that the myriad of snapping shrimp and fish vocalizations guide organisms (e.g. larva, fish, and marine mammals) into and within the estuary. Recent laboratory and field playback experiments conducted in St. Johns, US Virgin Islands and Pamlico Sound, North Carolina, USA have reported that larva utilize sound cues to find coral / oyster reefs based on the biogenic sound production of organisms occupying these habitats [35–37]. The May River is a salt marsh estuary bordered by extensive patches of smooth cordgrass *Spartina alterniflora* and oyster reefs comprised of the eastern oyster *Crassostrea virginica*. Passive acoustic recorders were placed on the bottom, close to the sides of the estuary, where live and dead oyster patches were common. Habitat-specific sound characteristics may reflect an important selection cue in driving settlement and recruitment patterns in marine communities, leading to higher biodiversity and potentially healthier habitats [36, 38].

While our study investigated temporal changes of the soundscape and diversity over a relatively short timeframe (~ six years), this approach provides a blueprint for implementation over longer time scales [3]. Listening to soundscapes has the potential to provide insight into the response and resiliency of individual species and their behaviors. Integrating long-term soundscape characterization into coastal marine observatory networks would be powerful because of its utility in providing acoustic behavior measurements at multiple levels of biological complexity (i.e. from snapping shrimp to fish to marine mammals) at time scales that range from minutes to years. This approach allows us to eavesdrop on key behaviors that can

change rapidly or gradually in response to environmental changes and human use; thus, it has potential to provide a measure of resilience or shifting baselines in a globally changing environment.

## Materials and methods

### Study area

The May River (32˚12'49"N, 80˚52'23"W) is located inland of the southern SC coast (Fig 5). This large subtidal river is ~22 km long and ~0.01 km wide near the source, and ~1 km wide at the mouth. The river is bordered by extensive patches of smooth cordgrass and oyster reefs (i.e. eastern oyster) with the town of Bluffton on the north-eastern side and Hilton Head Island at the mouth of the river. Water depth ranges from ~3 to ~7 m near the source and from ~4 to ~18 m near the mouth depending upon large semidiurnal tides. This area experiences a humid subtropical climate with hot summers and mild winters.

### Data collection and analysis

We deployed DSG-Ocean recorders (Loggerhead Instruments, Sarasota, FL, USA), water level and temperature loggers (HOBO 100-Foot Depth Water Level Data Logger U20-001-02-Ti and HOBO Water Temperature Pro v2 U22-001, Onset Computer Corporation, Bourne, MA, USA) at stations 9M, 14M, and 37M between February 2013 and December 2018 following methods previously described (Fig 5) [28]. Recorders collected sound samples for 2 min every 20 min at a sampling rate of 80 kHz over 22 deployments that were approximately 90 days long.

We determined the root mean square (rms) SPL for the entire data set (i.e. each 2 min wav file every 20 min; 392,106 files for all 3 stations) for high (i.e. 7000–40,000 Hz), low (i.e. 50–1200 Hz), and broadband (i.e. 1–40,000 Hz) frequencies using custom scripts created in MATLAB R2017b (MathWorks, Inc., Natick, MA, USA). We chose these ranges based on previous studies that revealed specific call frequencies for black drum (70–90 Hz), silver perch (1000–1280 Hz), oyster toadfish (190–200 Hz), spotted seatrout (200–270 Hz), red drum (120–160 Hz), and snapping shrimp (50–40 kHz) [28]. In the May River estuary, previous studies have discovered that the soundscape is composed mainly of biological sounds (i.e. snapping shrimp, fish, and bottlenose dolphins), physical sounds (i.e. wave, wind, water flow, and rain), and anthropogenic noise (i.e. recreational boats) [18]. Broadband frequency SPL values reflected biological sounds (i.e. snapping shrimp, fish, and bottlenose dolphins), physical sounds (i.e. wave, wind, water flow, and rain), and anthropogenic noise (i.e. recreational boats) [18]. Low frequency SPLs included fish calls, the lower bandwidth of snapping shrimp snaps, bottlenose dolphin vocalizations (which were few and random), physical sounds, and anthropogenic noise [18]. High frequency SPLs included snapping shrimp snaps, high frequency vocalizations of bottlenose dolphins, physical sounds, and anthropogenic noise [18].

In order to decipher biological patterns from physicals sounds and noise, we subsampled the data set and manually analyzed files recorded on the hour (i.e. 130,702 files for all 3 stations). We flagged the files that contained biological sounds (i.e. snapping shrimp, fish, and bottlenose dolphin), physical sounds (i.e. wave, wind, water flow, and rain), and anthropogenic noise (i.e. recreational boats) [18]. Then, we removed the files that contained physical sounds and anthropogenic noise from the subsampled data set and reanalyzed high, low, and broadband SPLs. This approach ensured that the patterns observed where of biological nature. We created heat maps in MATLAB R2017b using the entire data set representing all SPL values (i.e. 2 min on the hour including sounds of biological, physical, and anthropogenic origin) and the subsampled data set (i.e. 2 min on the hour including only sounds of biological origin).

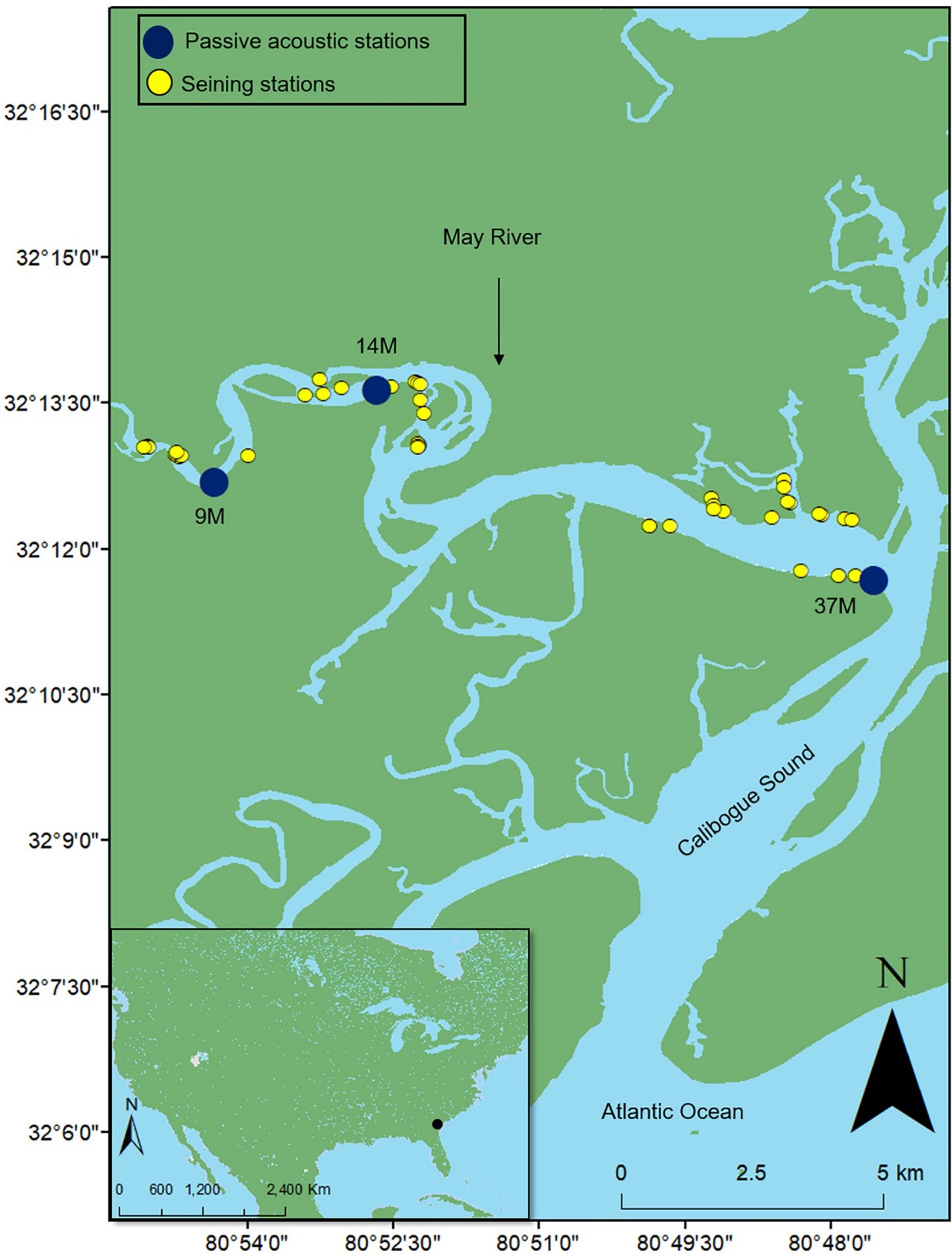

**Fig 5. Map of the May River, SC, USA.** Locations of stations 9M, 14M, and 37M that were acoustically monitored from February 2013 to December 2018 (blue) and seining stations monitored from May 2016 to December 2018 (yellow). (Inset) Location of the May River (black) in reference to the USA coast.

These SPL values were plotted versus date and time at each acoustic station (i.e. 9M, 14M, and 37M) with corresponding water temperature and daylight hours.

We performed invertebrate and fish sampling one to two times per month in the May River between 2016 and 2018 using a haul seine (i.e. seine width = 9.1 m, height = 1.2 m, and mesh diameter = 3 mm) and block nets (i.e. additional stationary seine nets to stop animals from escaping) at two to four locations per passive acoustic station (Fig 5). This sampling equated to six to twelve seines per month selected randomly from a list of sites. Seine sites included tidal pools (i.e. shallow pools of water created on the low tide), intertidal creeks (i.e. small secondary or tertiary creeks feeding from the main river accessible on the low tide), and sides of the river (i.e. stations located along the bank of the primary river). Before each seine, we recorded environmental parameters (i.e. water temperature, salinity, pH, and dissolved oxygen) using a YSI 556 Handheld Multiparameter Instrument (YSI Inc./Xylem Inc., Yellow Springs, OH, USA), and then we seined the area between the block nets. We measured the length and width of each seine for an area calculation. We transferred the catch into a live well, quantified the abundance of each species, and then released all the organisms at the original sampling location. Mortality of the organisms was low during the wintertime (< 5%) and moderate during summertime (~5 to 50%). We calculated average species richness per month based on the number of species in each seine standardized by the seine area. In addition, we calculated the average Shannon-Wiener diversity index divided by the area per month using package "vegan" in R version 3.4.2 (R Core Team, 2012) [39]. We calculated mean species abundance per month based on the abundance of all the species in each seine standardized by the seine area. This work was conducted under South Carolina Department of Natural Resources permit numbers 5135 and 5136 and IACUC protocol 2233-101181-022217.

## Statistical analyses

To assess the significance of specific factors in explaining variations in SPLs, we used package "Boruta", a wrapper algorithm based on the random forest algorithm in R [40–49]. Random forest models are non-parametric and do not require formal distribution assumptions. We used "permutation importance" rather than the default "mean decrease in impurity importance" to assess the importance of factors included in the model, since permutation importance is less biased of continuous and categorical variables with many levels [41–44]. In the final model, we set the specific parameters to: p = 0.01, mtry = 2, ntree = 200, nodesize = 5, and set.seed = 42 [45–47]. We included location, year, lunar phase, tide, day/night, temperature, day length, and rainfall as factors. We used four categories to differentiate the lunar and tidal cycle following methods previously described, and we used National Oceanic and Atmospheric Administration (NOAA) weather stations located close to the May River to obtain rainfall data for each day (S6 Table) [28]. Before applying the model, we tested the data for collinearity. Temperature, rainfall, and day length exhibited multi-collinearity that could bias the Boruta feature selection algorithm [47]. Hence, we created three different models for high, low, and broadband frequency SPL. In the first set of models, we included temperature as a factor, in the second set, we used day length, and in the third set, we used rainfall. Then, we compared the $R^2$ for each model that used a different factor, and we reported the models that best explained the variability of the data. We removed files that contained physical sounds and anthropogenic noise, and we followed the same approach to test variable importance. Then,

we compared the results of the random forest models applied to the two data sets (i.e. one data set that included biological sounds, physical sounds, and anthropogenic noise, and the second data set that included only biological sounds). If categorical variables were significant, we applied Dunnett-Tukey-Kramer pairwise multiple comparison tests adjusted for unequal variances and unequal sample sizes with 95% confidence levels using package "DTK" in R to determine whether group means were significantly different from each other [50].

To identify significant changes in high, low, and broadband SPLs values during the springtime, we applied a Bayesian change point analysis using package "bcp" in R [51, 52] to the dataset that excluded all physical sounds and anthropogenic noise. Bayesian change point analysis calculates the posterior probability (PP) for any given point in the time series that has an abrupt change. We defined a probability as significant when the change between the following and previous data point was $\geq$ 50% [8]. For each year, we reported the mean spring water temperature, date and day of year of the first significant PP, day anomaly (i.e. day of first significant PP for each year minus the six year average of days with the first PP), and the value of the first significant PP at each station. We defined spring as the time between the astronomical vernal equinox and the summer solstice.

We tested for normality by investigating the distribution of the residuals, and center and dispersion of monthly averages of species richness, or Shannon-Wiener diversity index, and species abundance. We performed Pearson's correlations between monthly averages of species richness (or Shannon-Wiener diversity index and species abundance) and monthly averages of high, low, and broadband frequency SPLs for all stations combined. In addition, using linear regression, we tested the relationship between monthly averages of temperature as the independent variable and corresponding monthly averages of species richness (or Shannon-Wiener diversity index and species abundance) as the dependent variable. Correlations and regressions were performed in MATLAB using two-sided hypothesis tests with a significance level of 0.05.

## Supporting information

**S1 Fig. Time series of broadband frequency Sound Pressure Levels (SPLs) from 2013 until 2018.** Heat maps represent temporal and spatial patterns of broadband (i.e. 1–40,000 Hz) frequency SPLs reflecting all physical sounds, biological sounds, and anthropogenic noise at stations (A) 9M, (B) 14M, and (C) 37M in the May River. Time is shown between noon and noon of the next day. Gaps in data = white, temperature = black line, and daylight hours = dotted line. Green dots indicate first posterior probability (PP) $\geq$ 0.5 detected during springtime. At station (C) 37M first PP was not calculated for spring 2017 due to missing acoustic data. (TIF)

**S2 Fig. Time series of high frequency Sound Pressure Levels (SPLs) from 2013 until 2018 with physical sounds and anthropogenic noise files removed.** Heat maps represent temporal and spatial patterns of high (i.e. 7000–40,000 Hz) frequency SPLs reflecting snapping shrimp acoustic activity at stations (A) 9M, (B) 14M, and (C) 37M in the May River. Time is shown between noon and noon of the next day. Gaps in data = gray, files with physical sounds and anthropogenic noise removed = white, temperature = black line, and daylight hours = dotted line. Green dots indicate first posterior probability (PP) $\geq$ 0.5 detected during springtime. At station (C) 37M first PP was not calculated for spring 2017 due to missing acoustic data. (TIF)

**S3 Fig. Time series of low frequency Sound Pressure Levels (SPLs) from 2013 until 2018 with physical sounds and anthropogenic noise files removed.** Heat maps represent temporal

and spatial patterns of low (i.e. 50–1200 Hz) frequency SPLs reflecting fish and lower portion of snapping shrimp acoustic activity at stations (A) 9M, (B) 14M, and (C) 37M in the May River. Time is shown between noon and noon of the next day. Gaps in data = gray, files with physical sounds and anthropogenic noise removed = white, temperature = black line, and day-light hours = dotted line. Green dots indicate first posterior probability (PP) ≥ 0.5 detected during springtime. At station (C) 37M first PP was not calculated for spring 2017 due to missing acoustic data.
(TIF)

**S4 Fig. Time series of broadband frequency Sound Pressure Levels (SPLs) from 2013 until 2018 with physical sounds and anthropogenic noise files removed.** Heat maps represent temporal and spatial patterns of broadband (i.e. 1–40,000 Hz) frequency SPLs reflecting all biological activity at stations (A) 9M, (B) 14M, and (C) 37M in the May River. Time is shown between noon and noon of the next day. Gaps in data = gray, files with physical sounds and anthropogenic noise removed = white, temperature = black line, and daylight hours = dotted line. Green dots indicate first posterior probability (PP) ≥ 0.5 detected during springtime. At station (C) 37M first PP was not calculated for spring 2017 due to missing acoustic data.
(TIF)

**S5 Fig. Estimated mean high frequency Sound Pressure Level (SPL) with Posterior Probability (PP) of change at station 9M.** Posterior probability of change during springtime (light blue line) with corresponding estimated mean high (7000–40,000 Hz) frequency SPL (dark blue line) and corresponding water temperature (red line) in years (A) 2013, (B) 2014, (C) 2015, (D) 2016, (E) 2017, and (F) 2018. Stars indicate first positive (i.e. detected change in estimated mean SPL due to an increase not a decrease in SPL values) PP ≥ 0.5.
(TIF)

**S6 Fig. Estimated mean low frequency sound pressure level with Posterior Probability (PP) of change at station 9M.** Probability of change during springtime (light blue line) with corresponding estimated mean low (50–1200 Hz) frequency SPL (dark blue line) and corresponding water temperature (red line) in years (A) 2013, (B) 2014, (C) 2015, (D) 2016, (E) 2017, and (F) 2018. Stars indicate first positive (i.e. detected change in estimated mean SPL due to an increase not a decrease in SPL values) PP ≥ 0.5.
(TIF)

**S7 Fig. Estimated mean broadband frequency sound pressure level with Posterior Probability (PP) of change at station 9M.** Probability of change during springtime (dark blue line) with corresponding estimated mean broadband (1–40,000 Hz) frequency SPL (light blue line) and corresponding water temperature (red line) in years (A) 2013, (B) 2014, (C) 2015, (D) 2016, (E) 2017, and (F) 2018. Stars indicate first positive (i.e. detected change in estimated mean SPL due to an increase not a decrease in SPL values) PP ≥ 0.5.
(TIF)

**S8 Fig. Estimated mean high frequency sound pressure level with Posterior Probability (PP) of change at station 14M.** Probability of change during springtime (light blue line) with corresponding estimated mean high (7000–40,000 Hz) frequency SPL (dark blue line) and corresponding water temperature (red line) in years (A) 2013, (B) 2014, (C) 2015, (D) 2016, (E) 2017, and (F) 2018. Stars indicate first positive (i.e. detected change in estimated mean SPL due to an increase not a decrease in SPL values) PP ≥ 0.5.
(TIF)

**S9 Fig. Estimated mean low frequency sound pressure level with Posterior Probability (PP) of change at station 14M.** Probability of change during springtime (light blue line) with corresponding estimated mean low (50–1200 Hz) frequency SPL (dark blue line) and corresponding water temperature (red line) in years (A) 2013, (B) 2014, (C) 2015, (D) 2016, (E) 2017, and (F) 2018. Stars indicate first positive (i.e. detected change in estimated mean SPL due to an increase not a decrease in SPL values) PP $\geq$ 0.5.
(TIF)

**S10 Fig. Estimated mean broadband frequency sound pressure level with Posterior Probability (PP) of change at station 14M.** Probability of change during springtime (light blue line) with corresponding estimated mean (1–40,000 Hz) broadband frequency SPL (dark blue line) and corresponding water temperature (red line) in years (A) 2013, (B) 2014, (C) 2015, (D) 2016, (E) 2017, and (F) 2018. Stars indicate first positive (i.e. detected change in estimated mean SPL due to an increase not a decrease in SPL values) PP $\geq$ 0.5.
(TIF)

**S11 Fig. Estimated mean high frequency sound pressure level with Posterior Probability (PP) of change at station 37M.** Probability of change during springtime (light blue line) with corresponding estimated mean high (7000–40,000 Hz) frequency SPL (dark blue line) and corresponding water temperature (red line) in years (A) 2013, (B) 2014, (C) 2015, (D) 2016, (E) 2017, and (F) 2018. Stars indicate first positive (i.e. detected change in estimated mean SPL due to an increase not a decrease in SPL values) PP $\geq$ 0.5. Gray box = no data.
(TIF)

**S12 Fig. Estimated mean low frequency sound pressure level with Posterior Probability (PP) of change at station 37M.** Probability of change during springtime (light blue line) with corresponding estimated mean low (50–1200 Hz) frequency SPL (dark blue line) and corresponding water temperature (red line) in years (A) 2013, (B) 2014, (C) 2015, (D) 2016, (E) 2017, and (F) 2018. Stars indicate first positive (i.e. detected change in estimated mean SPL due to an increase not a decrease in SPL values) PP $\geq$ 0.5. Gray box = no data.
(TIF)

**S13 Fig. Estimated mean broadband frequency sound pressure level with Posterior Probability (PP) of change at station 37M.** Probability of change during springtime (light blue line) with corresponding estimated mean broadband (1–40,000 Hz) frequency SPL (dark blue line) and corresponding water temperature (red line) in years (A) 2013, (B) 2014, (C) 2015, (D) 2016, (E) 2017, and (F) 2018. Stars indicate first positive (i.e. detected change in estimated mean SPL due to an increase not a decrease in SPL values) PP $\geq$ 0.5. Gray box = no data.
(TIF)

**S14 Fig. Time series of broadband frequency Sound Pressure Levels (SPLs) (i.e. with physical sounds and anthropogenic noise files removed), species richness, and abundance from 2016 until 2018.** Heat maps represent temporal and spatial patterns of broadband (1–40,000 Hz) frequency SPLs with corresponding species richness (black line), species abundance (blue dotted line), and temperature (red line) at stations (A) 9M, (B) 14M, and (C) 37M. Files with physical sounds and anthropogenic noise removed = white and no data = gray box.
(TIF)

**S15 Fig. Correlation and regression analysis of species richness and Shannon-Wiener diversity index, sound pressure levels, and temperature.** Pearson's correlation between monthly averages of species richness and monthly averages of sound pressure levels (SPLs) in (A) high (7000–40,000 Hz), (B) low (50–1200 Hz), and (C) broadband (1–40,000 Hz)

frequency ranges. Pearson's correlation (r) between monthly averages of Shannon-Wiener diversity index and monthly averages SPLs in (E) high (7000–40,000 Hz), (F) low (50–1200 Hz), and (G) broadband (1–40,000 Hz) frequency ranges. Linear regression between monthly averages of temperature as the independent variable and monthly averages of (D) species richness and (H) Shannon-Wiener diversity index as dependent variables between 2016 and 2018 at all stations combined. For correlations N = 86 and for regression N = 90.
(TIF)

**S16 Fig. Correlation and regression analysis of species abundance, sound pressure levels, and temperature.** Pearson's correlation between monthly averages of species richness and monthly averages of sound pressure levels (SPLs) in (A) high (7000–40,000 Hz), (B) low (50–1200 Hz), and (C) broadband (1–40,000 Hz) frequency ranges. Linear regression between monthly averages of temperature as the independent variable and monthly averages of (D) species abundance between 2016 and 2018 at all stations combined. For correlations N = 86 and for regression N = 90.
(TIF)

**S1 Table. Significance of specific variables on sound pressure levels.** Results of the Boruta, a wrapper algorithm based on random forest, that tested the significance of specific variables on high (7000–40000 Hz), low (50–1200 Hz), and broadband (1–40000 Hz) frequency sound pressure levels (SPLs). Decision was confirmed important at p < 0.01; N = 130,679.
(XLSX)

**S2 Table. Significance of specific variables on sound pressure levels with files that contained physical sounds and anthropogenic noise removed.** Results of the of the Boruta, a wrapper algorithm based on random forest, that tested the significance of specific variables on high (7000–40000 Hz), low (50–1200 Hz), and broadband (1–40000 Hz) frequency sound pressure levels (SPLs). Decision was confirmed important at p < 0.01; N = 130,679.
(XLSX)

**S3 Table. Results of first positive Posterior Probability of change (PP).** Year, station, mean spring water temperature, date, and day of year of first PP of change ≥ 0.5 with corresponding day anomaly, and value of PP for high (7000–40000 Hz) frequency sound pressure levels (SPLs).
(XLSX)

**S4 Table. Results of first positive Posterior Probability of change (PP).** Year, station, mean spring water temperature, date, and day of year of first PP of change ≥ 0.5 with corresponding day anomaly, and value of PP for low (50–1200 Hz) frequency sound pressure levels (SPLs).
(XLSX)

**S5 Table. List of species that were caught and quantified during seining conducted one or two times per month in close proximity to passive acoustic stations in the May River, SC.**
(XLSX)

**S6 Table. Locations of National Oceanic and Atmospheric Administration (NOAA) weather stations located close to the May River, SC.** Stations were used to obtain rainfall data for each day. Data were obtained from: https://www.ncdc.noaa.gov/cdo-web/search.
(XLSX)

**S1 Data.**
(XLSX)

**S2 Data.**
(XLSX)

## Acknowledgments

We thank Bob and Lee Brewer of May River Plantation for their support and for allowing us to use their community dock for our University of South Carolina Beaufort (USCB) research vessel. We also thank the following individuals for their help in collection and analysis of acoustic data: Alex Douglas, David Lusseau, Michael Powell, Matt Hoover, Rebecca Rawson, Steven Vega, Chris Kehrer, Jenna MacKinnon, Alishia Zyer, Andrea Berry, Mackenna Neuroth, Hannah Naylander-Asplin, Michaela Miller, Ashlee Seder, Somers Smott, Joshua Himes, Debra Albanese, Shaneel Bivek, Caleb Shedd, Austin Roller, Eva May, Allison Davis, Alyssa Marian, Jamileh Soueidan, and Jake Morgenstern.

## Author Contributions

**Conceptualization:** Eric W. Montie.

**Data curation:** Bradshaw McKinney, Claire Mueller, Eric W. Montie.

**Formal analysis:** Eric W. Montie.

**Funding acquisition:** Eric W. Montie.

**Investigation:** Agnieszka Monczak, Bradshaw McKinney, Claire Mueller, Eric W. Montie.

**Methodology:** Agnieszka Monczak, Bradshaw McKinney, Claire Mueller, Eric W. Montie.

**Project administration:** Eric W. Montie.

**Resources:** Eric W. Montie.

**Supervision:** Eric W. Montie.

**Validation:** Agnieszka Monczak, Eric W. Montie.

**Visualization:** Agnieszka Monczak, Eric W. Montie.

**Writing – original draft:** Agnieszka Monczak, Eric W. Montie.

**Writing – review & editing:** Agnieszka Monczak, Claire Mueller, Eric W. Montie.

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
