## [Decision Letter · Decision Letter 0]

1 Apr 2020

PONE-D-20-03292

What’s all that racket! Soundscapes, phenology, and biodiversity in estuaries

PLOS ONE

Dear Dr. Montie,

Thank you for submitting your manuscript to PLOS ONE. After careful consideration, we feel that it has merit but does not fully meet PLOS ONE’s publication criteria as it currently stands. Therefore, we invite you to submit a revised version of the manuscript that addresses the points raised during the review process.

I found this to be an interesting and well written study and both reviewers agreed. However, both reviewers have suggested some relatively minor changes and clarifications to the manuscript and I have provided some minor editorial comment that need to be addressed. I encourage the authors to consider all the comments provided when making their revisions.

We would appreciate receiving your revised manuscript by May 16 2020 11:59PM. To enhance the reproducibility of your results, we recommend that if applicable you deposit your laboratory protocols in protocols.io, where a protocol can be assigned its own identifier (DOI) such that it can be cited independently in the future. For instructions see: http://journals.plos.org/plosone/s/submission-guidelines#loc-laboratory-protocols

We look forward to receiving your revised manuscript.

Kind regards,

Heather M. Patterson, Ph.D.

Academic Editor

PLOS ONE

Journal Requirements:

Reviewers' comments:

Reviewer's Responses to Questions

**Comments to the Author**

1. Is the manuscript technically sound, and do the data support the conclusions?

Reviewer #1: Yes

Reviewer #2: Yes

2. Has the statistical analysis been performed appropriately and rigorously? 

Reviewer #1: Yes

Reviewer #2: Yes

3. Have the authors made all data underlying the findings in their manuscript fully available?

Reviewer #1: Yes

Reviewer #2: Yes

4. Is the manuscript presented in an intelligible fashion and written in standard English?

Reviewer #1: Yes

Reviewer #2: Yes

5. Review Comments to the Author

Reviewer #1: Review – PLOS One

Manuscript number: PONE-D-20-03292

What’s all that racket! Soundscapes, phenology, and biodiversity in estuaries

General Comments:

The overall objectives of this study were to; 1. Determine temporal patterns of low, high and broadband sound pressure levels over a six-year time span, 2. Determine how certain environmental factors influence SPLs, 3. Examine the phenology of acoustic activity of snapping shrimp and sound producing fish species, and 4. Determine temporal patterns of species diversity and abundance and examine how these indices correlate with the soundscape. This study provides a valuable addition to the growing underwater soundscape field, especially while adding the fairly novel addition of phenology.

Overall, I found this manuscript to be very well written, and an interesting and well-planned investigation. I applaud the 6-year time series in acoustic data collection and effort going into this sampling!

The authors need to discuss some potentially complicating factors such as abiotics, water depth and propagation distances etc. (see specific comments). These factors should be touched upon in the discussion.

Specific Comments:

Introduction

Line 64: Add ”,” between abundance & and.

Methods

Lines 237 – 238: Sound propagation and potential listening distance can vary between these depths. Would be good to acknowledge and briefly discuss these complexities.

Lines 253-254: In the future, also consider presenting medians, especially when capturing/including anthropogenic sound sources and in some cases loud chorusing events. Root mean square SPL can be more sensitive to changes in the right tail of the probability distribution, e.g., higher noise levels, therefore the magnitude of change would be larger.

Lines 258 – 262: Revise wording for clarity. Currently it reads as though you only subsampled the data for anthropogenic sounds, and therefore removed only a subset of this? Then you state you created heatmaps representing only biological sounds – this would not be the case if you subsampled only. Or did you only graph the subsampled data?

When noting representing biological sounds, what about abiotic sounds? I am guessing these are commonly occurring in these habitats, e.g. flow noise, wind and waves acting on the surface. Some of which would have been removed when filtering to 50 Hz, but not during broadband measurements. This should be addressed – potentially in methods or discussion.

Results

Line 83: [20] Referencing Material and Methods section is somewhat confusing.

What exactly are you referencing here? If not anther study, leave out of bibliography. If referencing this current studies method, say so.

Line 86: Reference to supplementary figures S2-S4. How are these related? As these graphs are with the anthropogenic noise removed. Revise for clarity.

Lines 94 – 96:

Lines 215 – 218: For a broader species view consider referencing early study showing habitat specific sound cues in promoting settlement in crustaceans:

• Stanley, J. A., Radford, C. A., Jeffs, A. G. Location, location, location – finding a suitable home in amongst the noise. Proceedings of the Royal Society B-Biological Sciences 279:1742, 3622 - 3631.

Figures

Very nice figures throughout, however, all figures within the main article are of very poor resolution. I am thinking this is just for the review manuscript to save on size, however if not, these need to be improved for better viewing as currently they are difficult to view and read in some places.

Figures 1 & 2: In Figure 1 the data gaps are in white (as per figure legend), in Figure 2 they are in Grey, although the figure legend notes white. This is probably just an oversite but stay consistent for clarity. Personally, I think grey illustrates it the best.

Figure 5: might be good if you could indicate what the blue and yellow icons were in a small inset of the figure for ease of viewing.

Reviewer #2: This study explored the phenology and biodiversity of a subtidal river using underwater acoustic recordings. The authors collected six-years of data, which demonstrated an increase in biotic sounds in years with warmer springs. This is a very interesting result, and is particularly relevant given global climate change.

However, the manuscript requires some clarifications. I have two key concerns:

Firstly, in the Methods it says that a subset of the data were manually reviewed to remove files containing anthropogenic noise - however, the Results then seem to alter between using the full dataset (i.e. containing anthropogenic noise) and the subset data (i.e. excluding anthropogenic noise). It would be beneficial to clarify throughout when you are using which dataset. Naming them could help with this.

Secondly, the subset data is split into low frequency (50 - 1200 Hz) and high frequency (7000 - 40,000 Hz). It then appears to be assumed that all sounds in the low-frequency category are caused by fish and those in the high-frequency category are caused by snapping shrimp. This is likely broadly true - but the assumption is not explicitly stated anywhere in the manuscript. Although the anthropogenic noise has been removed, there are still other sound sources that could be contributing to noise levels in these low- and high-frequency categories. For example, abiotic sounds (i.e. rain, wind, currents, bubbles, etc) can considerably alter ambient noise levels. See Marley et al (2016) Underwater sound in an urban estuarine river (DOI 10.1007/s40857-015-0038-z); here I found that wind strongly influenced the local soundscape. I appreciate that abiotic sounds can be difficult (impossible?) to remove; but as they are natural sounds, there is still the risk that they could be correlated with temperature (e.g. weather patterns, higher flow rates in the river, etc) and thus influence results. Rather than redoing any data review or analysis, I would like to see this assumption fully addressed in the Methods section.

Apart from these two key concerns, my comments are relatively minor suggestions. I have attached an annotated PDF of specific comments. In particular, I think it would be worth developing the Discussion further to highlight the relevance of this research in a changing climate. This is of particular importance given your own point in the Introduction - phenology is relatively overlooked in marine systems compared to terrestrial ones. So please do take the opportunity to elaborate further in the Discussion.

I look forward to seeing the final publication!

All the best,

Sarah Marley

6. PLOS authors have the option to publish the peer review history of their article (what does this mean?). If published, this will include your full peer review and any attached files.

Reviewer #1: No

Reviewer #2: Yes: Sarah Marley

---

## [Author Response · Author response to Decision Letter 0]

14 Jul 2020

The response to reviewers is provided in the cover letter

---

## [Editor Report · Decision Letter 1]

16 Jul 2020

What’s all that racket! Soundscapes, phenology, and biodiversity in estuaries

PONE-D-20-03292R1

Dear Dr. Montie,

We’re pleased to inform you that your manuscript has been judged scientifically suitable for publication and will be formally accepted for publication once it meets all outstanding technical requirements.

Kind regards,

Heather M. Patterson, Ph.D.

Academic Editor

PLOS ONE
---

## [Editor Report · Acceptance letter]

24 Aug 2020

PONE-D-20-03292R1 

What’s all that racket! Soundscapes, phenology, and biodiversity in estuaries 

Dear Dr. Montie:

I'm pleased to inform you that your manuscript has been deemed suitable for publication in PLOS ONE. Congratulations! Your manuscript is now with our production department. 

Kind regards, 

on behalf of

Dr. Heather M. Patterson 

Academic Editor

PLOS ONE